# Long term conservation of DNA at ambient temperature. Implications for DNA data storage

Delphine Coudy[1], Marthe Colotte[2], Aurélie Luis[1], Sophie Tuffet[1,2], Jacques Bonnet[1,3]*

**1** Laboratoire de Recherche et développement, Imagene Company, Pessac, France, **2** Imagene, plateforme de production, Genopole, Evry, France, **3** Université de Bordeaux, Institut Bergonié, INSERM, Bordeaux, France

* bonnet@imagene.eu

**Data Availability Statement:** All relevant data are within the manuscript and its Supporting Information files.

**Funding:** The authors received no specific funding for this work.

## Abstract

DNA conservation is central to many applications. This leads to an ever-increasing number of samples which are more and more difficult and costly to store or transport. A way to alleviate this problem is to develop procedures for storing samples at room temperature while maintaining their stability. A variety of commercial systems have been proposed but they fail to completely protect DNA from deleterious factors, mainly water. On the other side, Imagene company has developed a procedure for long-term conservation of biospecimen at room temperature based on the confinement of the samples under an anhydrous and anoxic atmosphere maintained inside hermetic capsules. The procedure has been validated by us and others for purified RNA, and for DNA in buffy coat or white blood cells lysates, but a precise determination of purified DNA stability is still lacking. We used the Arrhenius law to determine the DNA degradation rate at room temperature. We found that extrapolation to 25°C gave a degradation rate constant equivalent to about 1 cut/century/100 000 nucleotides, a stability several orders of magnitude larger than the current commercialized processes. Such a stability is fundamental for many applications such as the preservation of very large DNA molecules (particularly interesting in the context of genome sequencing) or oligonucleotides for DNA data storage. Capsules are also well suited for this latter application because of their high capacity. One can calculate that the 64 zettabytes of data produced in 2020 could be stored, standalone, for centuries, in about 20 kg of capsules.

## Introduction

Conservation of DNA, purified, in biospecimens or synthetic is a prerequisite to many applications, from biobanking, biodiversity preservation or molecular diagnostics to digital data storage (for reviews, see for instance [1–3]). This generates an ever-increasing number of samples which are more and more difficult and costly to store or transport. For reviews, see [4–6].

A way to alleviate, at least partially, this problem, is to develop procedures for storing samples at room temperature, allowing a standalone storage without energy costs. But, of course,

**Competing interests:** I have read the journal's policy and the authors of this manuscript have the following competing interests: D. Coudy, A. Luis, S. Tuffet and M. Colotte are employees of Imagene Company; S. Tuffet is CEO and shareholder of Imagene Company; J. Bonnet is a shareholder and a consultant of Imagene company.

this implies that DNA integrity must be maintained at room temperature which can be achieved by keeping DNA away from environmental degradation factors: water, oxygen, ozone, and other atmospheric pollutants [7–10], water being by far the most deleterious element.

Many systems, often based on dehydration, have been used for room temperature storage of purified DNA: freeze-drying [11], inclusion in soluble matrices including liposomes, polymers such as silk [12] or pullulan [13] or adsorption on solid supports such as natural or treated cellulose [14–16]. Other procedures use encapsulation in sol-gel-based silica [17,18] or in silica nanoparticles [19,20], inclusion in salts [21] or layered double hybrids [22], dissolution in deep eutectic solvents [23] or ionic liquids [24]. As none of these procedures can totally protect DNA from atmosphere or moisture, other ways have been proposed: protection under a gold film [25] or encapsulation under an inert atmosphere in hermetic stainless-steel capsules, the DNAshells™ (Imagene SA, France) [6,26,27].

To demonstrate the real efficacy of a given preservation procedure one must estimate the DNA rate of degradation at room temperature (here, 25°C) which is difficult because of the low degradation rate of dehydrated DNA in this condition. So, generally, one must rely on accelerated aging kinetics and extrapolation to room temperature by using Arrhenius equation. Such an approach has recently been used by Grass et al [28] and Organick et al [29], to compare some of these procedures in the context of DNA data storage. Among the tested procedures, DNA encapsulated in DNAshell™ did not give reliable rates of degradation because these were too low.

Here we report an Arrhenius analysis for purified DNA stored in DNAshell complementing these previous studies and exemplifying the high stability of DNA when stored under inert atmosphere.

## Material and methods

### DNA preparation

DNA was extracted from blood collected on EDTA, following the Puregene protocol (Gentra, Qiagen, Hilden Germany) and resuspended in 10 mM Tris-HCl, 1 mM EDTA, pH 8 and stored at 4°C.

### Ethics statement

The data regarding DNA stability presented in this study relate to projects that have been formally approved by the "*Comité de protection des personnes Sud Ouest et Outre Mer III*"°, including use of blood and blood-derived samples. "*L'Etablissement français du sang*" (EFS, France) is a French national establishment that is authorized to collect blood samples from adult volunteer donors for both therapeutic and non-therapeutic uses. The donations were collected in accordance with the French blood donation regulations and ethics and with the French Public Health Code (art L1221-1). Blood samples were anonymized according to the French Blood Center (EFS) procedure. Volunteer donors signed written informed consents before blood collection. EFS authorized Imagene to perform this study and provided de-identified blood and blood-derived samples for non-therapeutic use.

### DNA encapsulation

DNA encapsulation was realized as previously described [27,30]. Briefly, the DNA solutions (700 ng in 10 μL) were aliquoted in glass inserts fitted in open stainless-steel capsules (DNA-shells). The samples were dried under vacuum and left overnight in a glove box under an

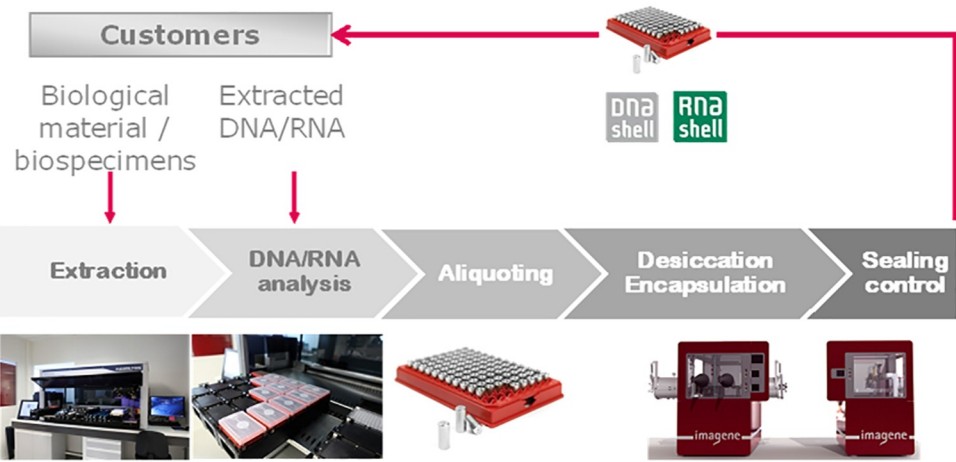

**Fig 1. Workflow of Imagene process.**

anoxic and anhydrous argon/helium atmosphere for further desiccation. Then, caps were added and sealed by laser welding. Finally, the DNAshells were checked for leakage by mass spectrometry [27,30]. Capsules are 18 mm x 7 mm weighing 1.3 g. They are made from deep drawing inox 304 with borosilicated glass inserts. This process is summarized in Fig 1.

One hundred and sixty capsules were produced (109 were used for gel electrophoresis and 47 for qPCR analysis).

## DNA accelerated degradation studies

DNAshells were heated in a Thermoblock at 100˚C, 110˚C,120˚C, 130˚C and 140˚C. At each time point of the kinetics, 2 or 3 capsules were retrieved, and stored at -20˚C. For analysis, the capsules were opened, the DNA samples were rehydrated with 20 µL of water. Half of the stored amount (350 ng) was immediately analyzed by electrophoresis. The remaining part of the samples were stored at -20˚C for qPCR.

## Measure of DNA degradation rates by qPCR with amplicons of two different sizes

Two amplicons of 1064 bp and a 93 bp of the TAF1L gene (*TATA-box binding protein associated factor 1 like*, Gene ID: 138474) were targeted. For both systems, PCR cycles were as follow: 10 min at 95˚C then 40 cycles of (15 s at 95˚C; 15 s at 60˚C; 60 s at 72˚C). The primers sequences were:

For-5' `agactcggacagcgaggaa`/ Rev-5' `cggagacacccagcatatca`
for the 1064 pb fragment and
For-5' `tgcaggcacttgagaacaac`/Rev-5' `aaccctgtcttgtccgaatg`
for the 93 pb.
They were produced by Eurogenetec, Les Ulis, France.

The runs were made on the CFX96 Touch Real-Time PCR Detection System (BIO-RAD LABORATORIES, INC).

After rehydration (with 20 µL of water), for each sample and each amplicon, we did a first 2/7 dilution then two ten-fold dilutions to estimate the PCR efficiency and construct the reference straight line. The diluted samples were analyzed independently and defined as "standards".

For the heated samples, for each time point, three capsules were taken and for each capsule qPCR determinations were done in triplicate. We used 8.35 ng aliquots for each qPCR determination.

To determine DNA recovery and degradation rates, we used a previously developed model [9] based on qPCR amplification of two amplicons (1 and 2) of different sizes ($L_1$ and $L_2$) to measure the number of cuts per nucleotide (or the probability of breakage at a given position). Assuming a random breakage mechanism, the probability of breakage at a given position is:

$$P_{cut} \ = \ 1 - e^{-kt}$$

and the probability of this position remaining unbroken is:

$$P_{uncut} \ = \ 1 - P_{cut} \ = \ e^{-kt}$$

From the model:

$$P_{uncut} \ = \ \frac{N_1}{N_2} e^{\frac{1}{L_1 - L_2}} \ = \ e^{-kt}$$

So, for each temperature, T, a graph of

$$\frac{N_1}{N_2} e^{\frac{1}{L_1 - L_2}}$$

versus time gave us $k_T$ by curve fitting.

This method is more reliable than one-sized qPCR because it is independent of the recovery.

Nevertheless, to determine the recovery of total genomic DNA, we used the formula:

$$N \ = \ \left( \frac{N_2^{L_1 - 1}}{N_1^{L_2 - 1}} \right)^{\frac{1}{L_2 - L_1}} \text{drawn from the same model}$$

## Results and discussion

The experimental strategy is summarized in the workflow shown in Fig 2.

### Measure of DNA degradation rates by qPCR with two different sizes amplicons

The samples were heated at 100°C, 110°C, 120°C, 130°C and 140°C for periods of time ranging from 2 min to 48 h. The Table 1 gives the number capsules used for each temperature and time point.

First, we ran electrophoresis as size controls to choose the time points corresponding to DNA sizes small enough ($< 8$ kb apparent size) to give significant values by qPCR. Indeed, when DNA size is too large, $N_1$ and $N_2$ not different enough. The gels are shown in **S1 File**.

The qPCR curves for all the experiments are given in **S2 File**.

**DNA recovery.**   From these curves we obtained the number of amplifiable copies of both amplicons TAF 93 ($N_2$) and TAF 1064 ($N_1$) for each temperature and each time point.

These results are presented in **S3 File**. From these values we could determine the total genomic DNA recovery as previously described [9] by using the formula:

$$N \ = \ \left( \frac{N_2^{L_1 - 1}}{N_1^{L_2 - 1}} \right)^{\frac{1}{L_2 - L_1}}$$

These recoveries are given in Table 2 and Fig 3.

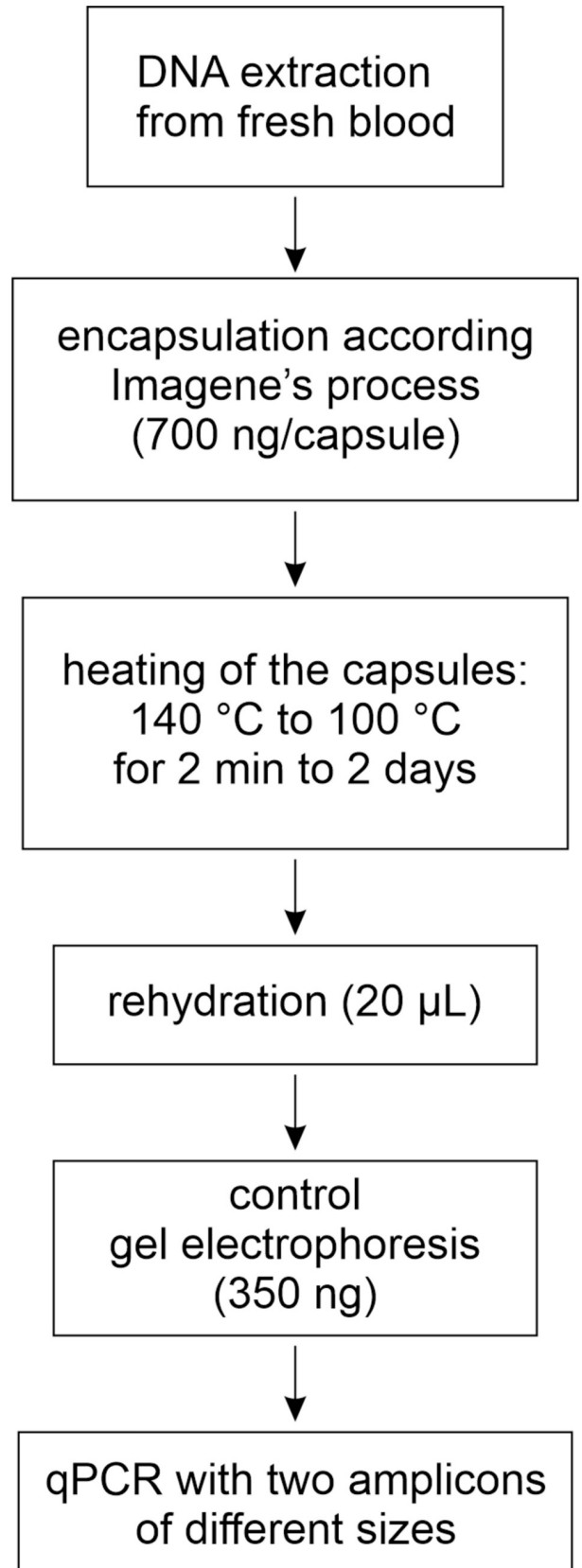

**Fig 2. Experimental strategy.**

**Table 1. Number of capsules used for each temperature and time point.**

| Temperature | Number of time points per kinetic | Number of capsules per time point | Total number of capsules |
|---|---|---|---|
| 140 | 3 | 3 | 9 |
| 130 | 4 | 2 | 8 |
| 120 | 3 | 3 | 9 |
| 110 | 3 | 3 | 9 |
| 100 | 4 | 3 | 12 |

It can be seen that the recovery of genomic DNA is very good in spite of some variability in the values. In particular, it does not seem to exist a decrease in recovery as a function of time and temperature.

As expected, the number amplifiable copies of the 1064 amplicon decreased over time while those of the 93 amplicon did not exhibit a significant decrease.

**Degradation kinetics of DNA stored in DNAshells.** For each time point, we calculated the proportion of intact nucleotide position, $P_{uncut}$, from the numbers of amplifiable copies of the 1064 pb and 93 pb amplicons. We plotted these values as a function of the degradation time to determine the degradation rate constants for each temperature by curve fitting (Fig 4).

Then, we plotted the logarithm of $k_T$ as a function of the reverse of the absolute temperature (T) (Fig 5).

This plot showed that the degradation rate followed the Arrhenius law with an activation energy of 197 kJ/mol. This is comparable to the 163 kJ/mol to 188kJ/mol previously found for desiccated plasmid DNA [7] and to about the 155 kJ/mol for DNA stored in silica nanoparticles, FTA paper or DNAstable matrix [28]. This is significantly higher than the 100 kJ/mol to 121 kJ/mol found for degradation of double strand DNA in solution (reviewed in [7]).

The Arrhenius law also made it possible to extrapolate the degradation rate at 25°C. This gave a degradation rate constant of $3.82 \times 10^{-15}$ cuts /s/nucleotide, equivalent to about 1 cut/century/100 000 nucleotides or 38 000 years of half-life for a 150-nucleotide long DNA fragment (we chose this size for an immediate comparison with the previous works [28,29]).

This allows to calculate the time necessary for a DNA molecule to degrade down to 25 nucleotides, the average length which is the current limit for the sequencing of degraded DNA [31] using the formula:

$$L_{max} \;=\; \frac{1}{P_{cut}} \;=\; \frac{1}{1 - e^{-kt}}$$

This gives 1,070,000 years, provided the preservation conditions are maintained.

**Table 2. Genomic DNA recovery.**

| heating temperature (°C) | heating time | | | | | | | |
|---|---|---|---|---|---|---|---|---|
| | t1 | | t2 | | t3 | | t4 | |
| | mean recovery (%) | standard deviation | mean recovery (%) | standard deviation | mean recovery (%) | standard deviation | mean recovery (%) | standard deviation |
| 140 | 71% | 19 | 88% | 15 | 101% | 10 | | |
| 130 | 139% | 35 | 169% | 8 | 162% | 9 | | |
| 120 | 53% | 1 | 134% | 23 | 113% | 20 | | |
| 110 | 132% | 26 | 210% | 9 | 115% | 3 | | |
| 100 | 81% | 12 | 175% | 18 | 135% | 9 | 93% | 2 |

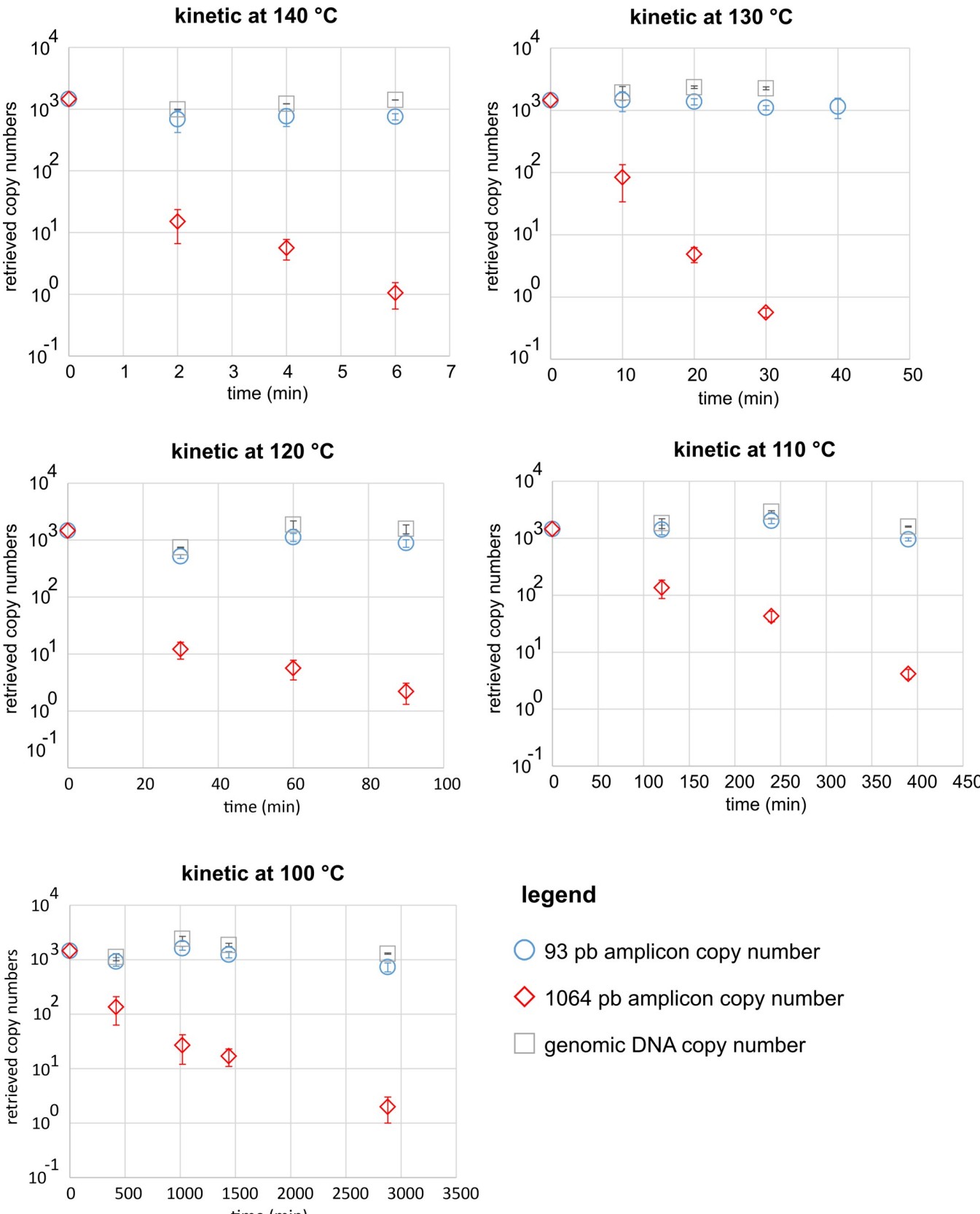

**Fig 3. Amplifiable copy numbers of 1064 amplicon, 93 amplicon and genomic DNA.**

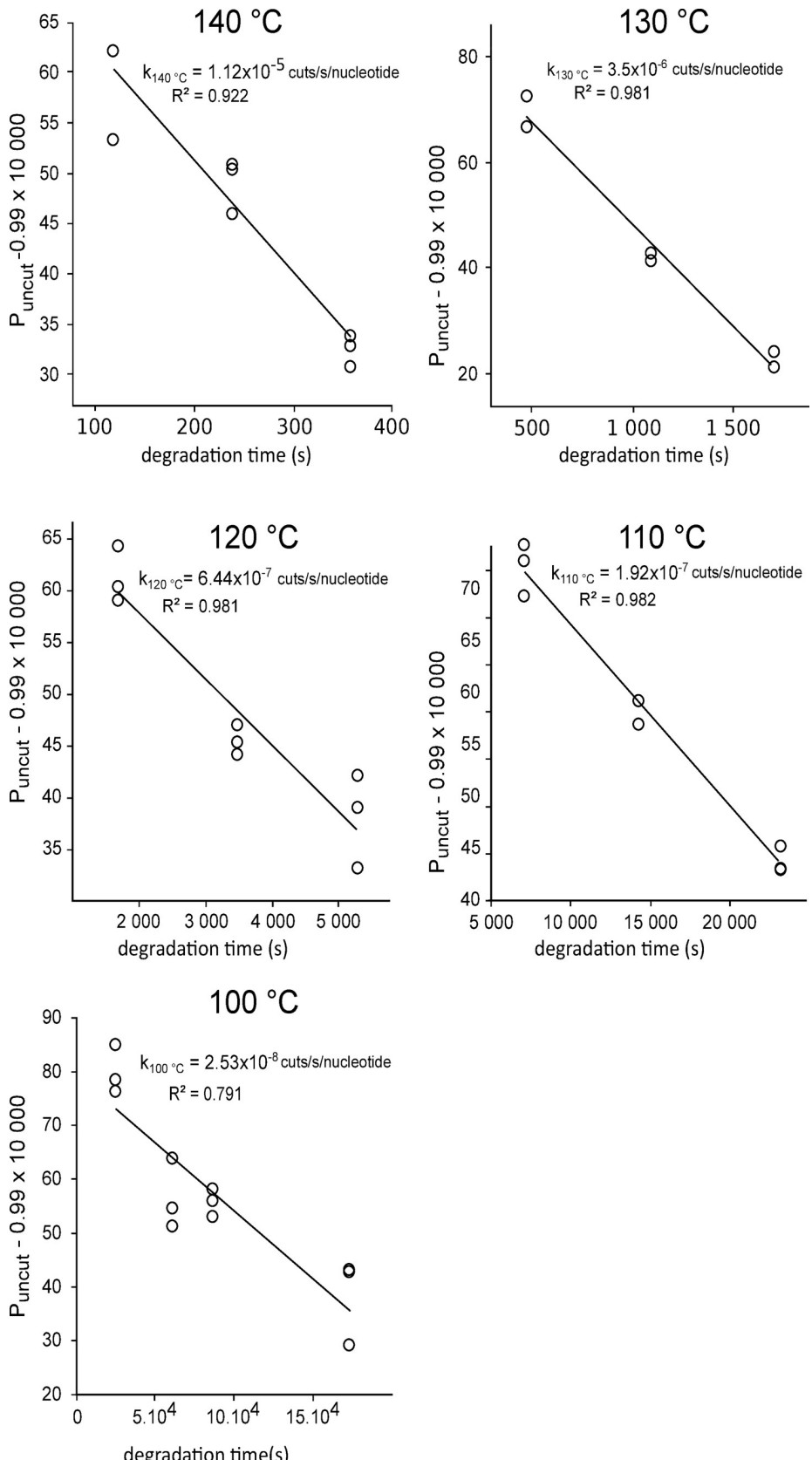

**Fig 4. Degradation kinetics of DNA stored in DNAshells.** The lines are the fit to the data points by Microsoft Excel software.

Fig 6 compares the half-lives of a 150 nucleotides long DNA fragment stored in various conditions at 25˚C.

So, it appears that the DNA stability at room temperature (25˚C) is over three orders of magnitude higher in DNAshells than in any of the other currently commercialized storage devices. This is to be expected because, first, FTA paper, trehalose or calcium phosphate leaves the DNA samples directly exposed to the atmosphere. Second, likewise, the matrices coating DNA: Gentegra, DNA stable and trehalose being water soluble cannot either protect the sample from moisture. Finally, silica nanoparticles, while affording protection from atmosphere, still contain a certain amount of water [28].

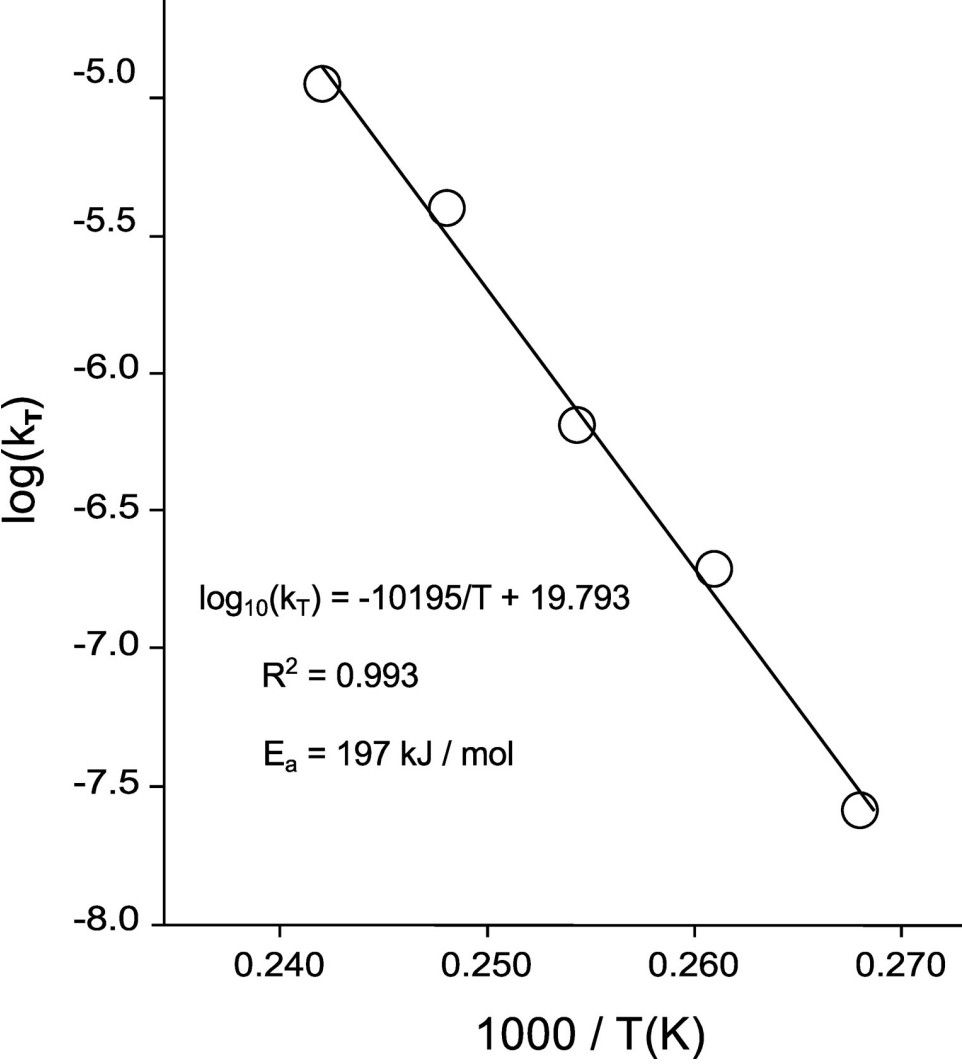

**Fig 5. Arrhenius plot for DNA degradation in DNAshells.** The degradation rate constants, k, were plotted as a function of the reverse of the absolute temperature T.

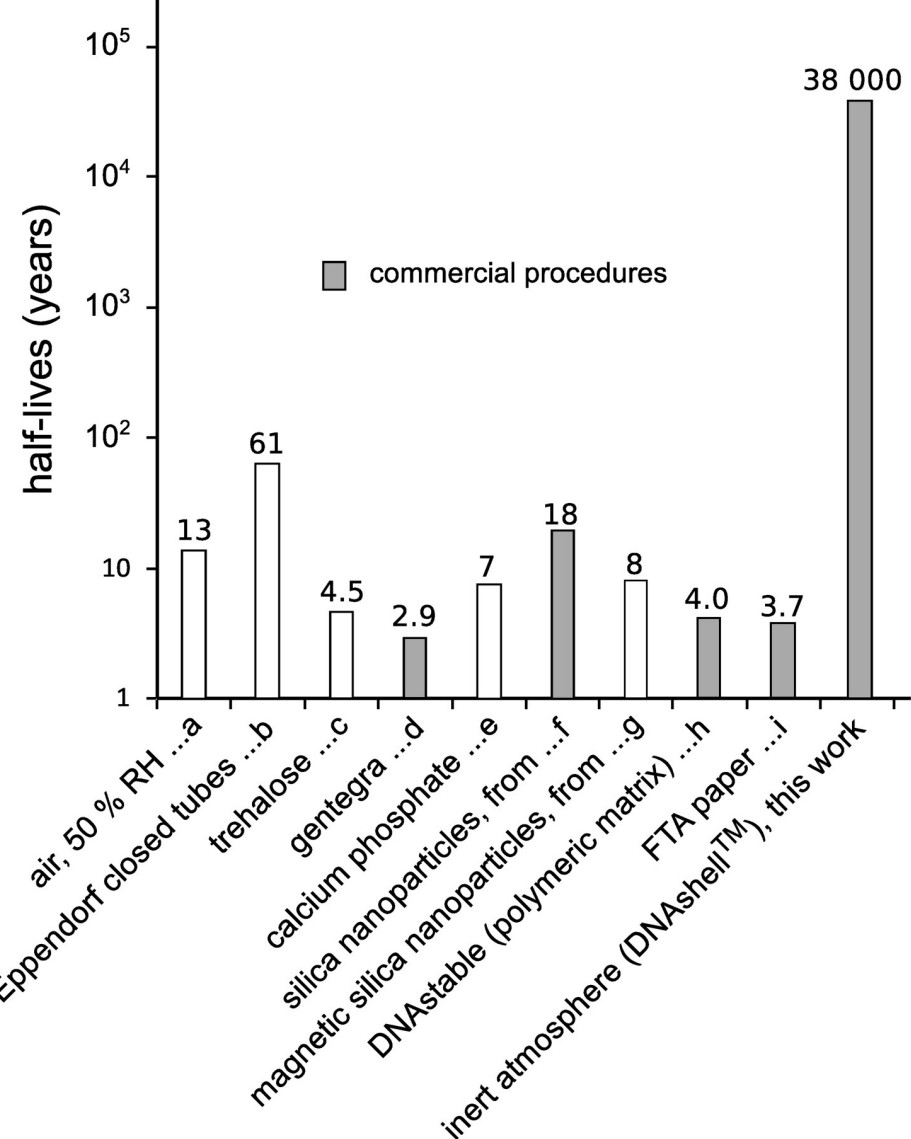

**Fig 6. Half-lives of a 150 nucleotides long DNA fragment stored in various conditions.** The half-life of a DNA sample left unprotected from the atmosphere at room temperature (a) or in an Eppendorf closed tube (b) has been calculated from our previous work [7]. The one of a sample encapsulated in silica nanoparticles (g), deposited on FTA card paper (i) or included in Biomatrica DNAstable (h) has been estimated from [28] (Fig 2B). The half-life of DNA dried with calcium phosphate (e) and encapsulated in magnetic silica nanoparticles (f) have been estimated, respectively from [29] (Fig 3B) and [19] (Fig 5) assuming an exponential decay and an activation energy of 155 kJ/mol. The half-life for DNA stored in Gentegra (d) or trehalose (c) was taken from [29] (Fig 2B). In grey: Current commercialized procedures.

It must be noticed that the experiments conducted here mainly detect chain breaks, so, other degradations events not preventing elongation by the polymerase could go undetected. More seriously, these undetected modifications could induce errors in the decoding step. However, this should not be a concern, first because DNA alterations are dependent on water

and much slower than depurination and chain breaks [32]. Second controls have been done by Organick et al ([29]) who sequenced DNA samples stored at 85˚C for 4 weeks. As a whole, they found that that the number of sequencing errors did not increase with storage and that these errors were stochastic and could be *"dealt with easily with various means of error correction such as Reed–Solomon codes"*.

As a conclusion, this procedure, allowing a standalone storage is well suited for long term preservation of DNA samples because of the high percentage of DNA retrieval and DNA stability. This is especially useful for the recently developing DNA data storage procedures. Our figures make it possible to give an estimation of the lifetime of the data stored that way. Indeed, according to a recent estimation by Organick et al [33], 10 is the lowest copy number necessary for a faithful data storage and retrieval. This means that if one starts with 20 copies, the data could faithfully be retrieved after 30 centuries of storage.

Another advantage is the volume of the capsule which can accommodate large amounts of DNA. With a 200 µL useful volume and a DNA density of 1.4 g/mL [34], a single capsule could store 0.28 g of DNA. According to [35] estimating at about 17 exabytes/g the data density in DNA, this corresponds to 4.76 exabytes of data per DNAshell, equivalent to $1.6 \times 10^{12}$ files (assuming an average file size of 3 MB). Of course, it may look difficult to recover a specific file among this mass of data, however, this seems possible as described recently by Tomek et al claiming that, by using a combination of N primers, it could be possible to select a given file in a population of $27\,999^{N}$ files [36].

So, according to these figures, the 64 zettabytes of data produced in 2020 [37] could theoretically be coded in 3 765 g of DNA which could be stored in 13,445 capsules packed in a suitcase weighing 21 kg.

This procedure could also allow the long-term room temperature preservation of very large DNA molecules which is particularly interesting in the context of genome sequencing, as a recent paper by Nurk et al described for the first time the sequencing of a complete human genome thanks to the use of very long DNA stretches [38].

## Supporting information

**S1 File. Gel electrophoresis.**
(DOCX)

**S2 File. qPCR curves.**
(PPTX)

**S3 File. qPCR results.**
(XLSX)

## Acknowledgments

We thank Régine Gandoin for editing the manuscript.

## Author Contributions

**Conceptualization:** Delphine Coudy, Marthe Colotte, Aurélie Luis, Sophie Tuffet, Jacques Bonnet.

**Data curation:** Delphine Coudy.

**Formal analysis:** Delphine Coudy, Marthe Colotte, Jacques Bonnet.

**Investigation:** Delphine Coudy, Marthe Colotte, Jacques Bonnet.

**Methodology:** Delphine Coudy, Marthe Colotte, Aurélie Luis, Jacques Bonnet.

**Project administration:** Delphine Coudy, Marthe Colotte.

**Supervision:** Sophie Tuffet, Jacques Bonnet.

**Validation:** Delphine Coudy, Marthe Colotte, Aurélie Luis, Sophie Tuffet, Jacques Bonnet.

**Visualization:** Delphine Coudy, Jacques Bonnet.

**Writing – original draft:** Sophie Tuffet, Jacques Bonnet.

**Writing – review & editing:** Delphine Coudy, Marthe Colotte, Aurélie Luis.

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
