## [Decision Letter · Decision Letter 0]

18 Aug 2021

PONE-D-21-23797

Long term conservation of DNA at ambient temperature. Implications for DNA data storage

PLOS ONE

Dear Dr. Bonnet,

Thank you for submitting your manuscript to PLOS ONE. After careful consideration, we have decided that your manuscript does not meet our criteria for publication and must therefore be rejected.

I am sorry that we cannot be more positive on this occasion, but hope that you appreciate the reasons for this decision.

Yours sincerely,

Mashallah Rezakazemi

Academic Editor

PLOS ONE

Reviewers' comments:

Reviewer's Responses to Questions

**Comments to the Author**

1. Is the manuscript technically sound, and do the data support the conclusions?

Reviewer #1: No

Reviewer #2: Yes

Reviewer #3: Yes

2. Has the statistical analysis been performed appropriately and rigorously? 

Reviewer #1: No

Reviewer #2: Yes

Reviewer #3: Yes

3. Have the authors made all data underlying the findings in their manuscript fully available?

Reviewer #1: No

Reviewer #2: Yes

Reviewer #3: No

4. Is the manuscript presented in an intelligible fashion and written in standard English?

Reviewer #1: Yes

Reviewer #2: Yes

Reviewer #3: Yes

5. Review Comments to the Author

Reviewer #1: Long term conservation of DNA at ambient temperature. Implications for DNA data storage by Delphine Coudy , Marthe Colotte , Aurélie Luis , Sophie Tuffet and Jacques Bonnet

Herein the authors demonstrate a technique to encapsulate DNA and claim its long term stability.

I have few comments and I strongly feel the study is not matured enough and lacks adequate merits to be published in Plos One.

1. After reading the abstract it is not clear how the authors have stabilized DNA, however after going through material and methods it is somewhat clear that the authors have dehydrated DNA aqueous solution and then stored in stain less steel capsules in very low concentration. This should be written in abstract as well.

2. The authors mentioned that the encapsulation process is described by them previously but have not provided any reference.

3. I don’t see any electrophoresis figures to ascertain the claim. It should be provided as supporting information.

4. The authors have heated the capsules, I believe the stability is due to the absence of moisture, stain less steel is not playing any role except stopping the heat transmission due to perhaps its insulating behaviour. In this case, what exactly is the advantage or role of stainless steel capsules? In any case if water somehow enters it can denature DNA.

5. The authors heat the steel capsules maximum at 140 deg C, DNA degrade in dry condition above 190 deg C.

6. Dimension of capsule shells should be provided, their supplier, type of steel used etc. may be provided.

7. Since authors have a noble aim to preserve the DNA samples under ambient conditions, the DNA should have been kept under ambient conditions after taking out from the shells and then carry out the stability study. This will help the use of DNA for material preparation where robust processing of the building block is desired.

8. DNA Stability study using some other technique should be attempted.

Reviewer #2: The manuscript presents results for the long-term conservation of DNA at ambient temperature. The study demonstrates an innovative way of DNA at ambient temperature. From my point of view, this is a well-performed study, which provides a rationale for further validation studies. However, the paper requires minor revision to become suitable for the readership.

(from the DNAshells™ to the result)

Materials and Methods

An illustration of the workflow would be helpful (samples -> conservation method -> degradation -> qPCR -> result).

Also, a schematic illustration of DNA encapsulation process would be helpful.

How many samples were used for which conservation method? -> A table would certainly be helpful here.

How much DNA was used?

qPCR: Which device was used with which settings?

Results:

qPCR curves should be demonstrated.

DNA recovery and qPCR results should be presented in a table for all conservation methods.

Reviewer #3: I recommend this work for publication, pending minor clarifications. In this work, the authors took two amplicons of different length and aged them at temperatures ranging from 100C to 140C for various lengths of time. The amount of full length product of each sample was then quantified via qPCR, and the extrapolated half life was found to be orders of magnitude longer than previously published preservation methods. The work was easy to understand and clearly described, I have only minor clarifications that should be addressed.

Minor clarifications:

I could not find any mention in the text about what the time points were. Looking at Figure 1, it appears the time points varied from hundredths of a second (i.e., 20x10^-3) to 25 hours (1500 seconds). Are the x-axis mislabeled in the plots for temperatures 100, 110 and 120? A sample will not have had time to come up to a uniform temperature with short amounts of time elapsed, I wonder if the authors meant 10^3?

Could you label the y-axis of Fig. 3?

“Moreover, Organick et al sequenced the DNA samples stored at 85 °C for 4 weeks without noticing any increase in error rates while there is a significant number of chain breaks [29].”

Organick et al [29] did not sequence DNA with chain breaks, so while they found no increase in error rates, I do not think we can claim that DNA that doesn’t break has a similar no increase in error rate. Based on the ambiguous wording of the above sentence, I would rephrase the sentence you wrote to make your claim more clear.

“With a 200 μL useful volume and a DNA density of 1.4 [34], a single capsule could store 0.28 g of DNA. According to [35] estimating at about 17 exabytes/g the data density in DNA, this corresponds to 4.76 exabytes of data per DNAshell, equivalent to 1.6x10^12 files (assuming an average file size of 3 Mo).”

Could you put units on the “1.4” value given, and I believe you mean “3 MB” instead of “3 Mo”?

The Data Availability statement states that all data are fully available, where is the qPCR data located? I wished to look at the data from the two amplicons of different length since neither the figures nor text do not explicitly show how breakage rates varied between the two lengths.

Minor typos:

The sentence: “This implies to keep DNA away from environmental 58

degradation factors, water, oxygen, ozone, and other atmospheric pollutants[7,8,9,10], water 59 being by far the most deleterious element” is incomplete, I’m not sure exactly what the authors are intending to communicate here.

6. PLOS authors have the option to publish the peer review history of their article (what does this mean?). If published, this will include your full peer review and any attached files.

Reviewer #1: No

Reviewer #2: No

Reviewer #3: No

- - - - -

---

## [Author Response · Author response to Decision Letter 0]

30 Sep 2021

Response to Reviewers 

August 21 2021

Reviewer #1: Long term conservation of DNA at ambient temperature. Implications for DNA data storage by Delphine Coudy , Marthe Colotte , Aurélie Luis , Sophie Tuffet and Jacques Bonnet

Herein the authors demonstrate a technique to encapsulate DNA and claim its long term stability.

I have few comments and I strongly feel the study is not matured enough and lacks adequate merits to be published in Plos One.

1. After reading the abstract it is not clear how the authors have stabilized DNA, however after going through material and methods it is somewhat clear that the authors have dehydrated DNA aqueous solution and then stored in stain less steel capsules in very low concentration. This should be written in abstract as well.

answer

 In the abstract we stated "…procedure for long-term conservation of biospecimen at room temperature based on the confinement of the samples under an anhydrous and anoxic atmosphere maintained inside hermetic capsules…" The principle of the procedure is quite simple, consisting essentially in protecting the samples in an airtight capsule. There is no need to add further details.

2. The authors mentioned that the encapsulation process is described by them previously but have not provided any reference.

answer

We did provide two references at the end of the paragraph. However, for more clarity, in the revised version we have also added them after the first sentence (Line 98).

3. I don’t see any electrophoresis figures to ascertain the claim. It should be provided as supporting information.

answer

Gel electrophoresis were run only as size controls to choose the time points corresponding to DNA sizes small enough (< 8 kb apparent size) to give significant values by qPCR.

Gel electrophoresis are given in Supporting information file S1 (Line 164).

4. The authors have heated the capsules, I believe the stability is due to the absence of moisture, stain less steel is not playing any role except stopping the heat transmission due to perhaps its insulating behavior. In this case, what exactly is the advantage or role of stainless steel capsules? In any case if water somehow enters it can denature DNA.

answer

It is clear, from our introduction, that moisture is indeed the main problem for DNA preservation, and this point is clearly stated in the papers given in reference, for instance: 

27. Colotte M, Coudy D, Tuffet S, Bonnet J. Adverse Effect of Air Exposure on the Stability of DNA Stored at Room Temperature. Biopreserv Biobank. 2011; 9(1): 47-50. doi: 10.1089/bio.2010.0028.

It is also clear that the role of the capsule is precisely to prevent water from reaching the DNA (using metallic container and welding being the only way to have an absolute hermeticity) and in no case to constitute an insulation barrier (which a metal cannot do).

Water does not denature DNA but contributes to its degradation. 

5. The authors heat the steel capsules maximum at 140 deg C, DNA degrade in dry condition above 190 deg C.

answer

If this means that dry DNA cannot degrade below 190-degree C, it cannot be true: DNA, even dry, degrades at any temperature, the degradation rates just increasing with temperature.

6. Dimension of capsule shells should be provided, their supplier, type of steel used etc. may be provided.

answer

Details about the characteristics of the capsules have been added (Line 103).

7. Since authors have a noble aim to preserve the DNA samples under ambient conditions, the DNA should have been kept under ambient conditions after taking out from the shells and then carry out the stability study. This will help the use of DNA for material preparation where robust processing of the building block is desired.

answer

Room temperature degradation studies of DNA, protected or unprotected, have previously been reported by us or others, for instance in the above cited paper (ref [27]). Taking the DNA out of the shell is only for analysis purposes, at the end of the room temperature storage or ageing period (potentially stopped by cold storage during the remaining time of the kinetics). The DNA had be used quickly after rehydration to take into account only the events of the analyzed period. 

8. DNA Stability study using some other technique should be attempted.

answer

There is a whole literature describing such techniques, in particular: [29]. Organick L, Nguyen BH, McAmis R, Chen WD, Kohll AX, Ang SD, et al. An Empirical Comparison of Preservation Methods for Synthetic DNA Data Storage. Small Methods. 2021; 5. The aim of our paper is precisely to complement these works.

Reviewer #2: The manuscript presents results for the long-term conservation of DNA at ambient temperature. The study demonstrates an innovative way of DNA at ambient temperature. From my point of view, this is a well-performed study, which provides a rationale for further validation studies. However, the paper requires minor revision to become suitable for the readership.

(from the DNAshells™ to the result)

Materials and Methods

1- An illustration of the workflow would be helpful (samples -> conservation method -> degradation -> qPCR -> result).

answer

This workflow has been added as Fig 2. at the line 152.

2- Also, a schematic illustration of DNA encapsulation process would be helpful.

answer

A workflow summarizing the encapsulation process has been added as Fig 1., line 105.

3- How many samples were used for which conservation method? -> A table would certainly be helpful here.

answer

A table (table 1) has been added (Line 158).

4- How much DNA was used?

answer

This is indicated in the experimental strategy (Fig 2 and line 111 and 130).

5- qPCR: Which device was used with which settings?

answer

The PCR apparatus used is specified line 124.

6- Results:

qPCR curves should be demonstrated.

answer

qPCR curves have been added in Supporting Information File S2- qPCR curves, line 164.

7- DNA recovery and qPCR results should be presented in a table for all conservation methods.

answer

A table (table 2) has been added to show DNA recovery (line 172).

A table presenting the qPCR results are presented in Supporting Information File S3, qPCR results (line 163). Fig 3. has been added to visualize theses results (line 174).

Reviewer #3: I recommend this work for publication, pending minor clarifications. In this work, the authors took two amplicons of different length and aged them at temperatures ranging from 100C to 140C for various lengths of time. The amount of full length product of each sample was then quantified via qPCR, and the extrapolated half life was found to be orders of magnitude longer than previously published preservation methods. The work was easy to understand and clearly described, I have only minor clarifications that should be addressed.

Minor clarifications:

1- I could not find any mention in the text about what the time points were. 

answer

An indication concerning the time points (ranging from 2 min to 48 h) has been added line 156. 

- Looking at Figure 1, it appears the time points varied from hundredths of a second (i.e., 20x10^-3) to 25 hours (1500 seconds). Are the x-axis mislabeled in the plots for temperatures 100, 110 and 120? A sample will not have had time to come up to a uniform temperature with short amounts of time elapsed, I wonder if the authors meant 10^3?

answer

We modified the time labels to make them clearer. The original Fig 1 is now Fig 3.

2- Could you label the y-axis of Fig. 3?

answer

The y-axis of Fig 3 (now Fig 5) has been labeled. 

3- “Moreover, Organick et al sequenced the DNA samples stored at 85 °C for 4 weeks without noticing any increase in error rates while there is a significant number of chain breaks [29].”

Organick et al [29] did not sequence DNA with chain breaks, so while they found no increase in error rates, I do not think we can claim that DNA that doesn’t break has a similar no increase in error rate. Based on the ambiguous wording of the above sentence, I would rephrase the sentence you wrote to make your claim more clear.

answer

The sentence has been rewritten.

4- “With a 200 μL useful volume and a DNA density of 1.4 [34], a single capsule could store 0.28 g of DNA. According to [35] estimating at about 17 exabytes/g the data density in DNA, this corresponds to 4.76 exabytes of data per DNAshell, equivalent to 1.6x10^12 files (assuming an average file size of 3 Mo).”

Could you put units on the “1.4” value given, 

answer

The units have been added (Line 246).

5- and I believe you mean “3 MB” instead of “3 Mo”?

answer

This has been corrected (Line 249).

The Data Availability statement states that all data are fully available, where is the qPCR data located? I wished to look at the data from the two amplicons of different length since neither the figures nor text do not explicitly show how breakage rates varied between the two lengths.

answer

We added qPCR data in Supporting Information File S3 qPCR results) giving, for each temperature and each time point the amplifiable copy numbers of both amplicons. These data indicate how the breakage rates "varied between the two lengths".

6- Minor typos:

The sentence: “This implies to keep DNA away from environmental 58

degradation factors, water, oxygen, ozone, and other atmospheric pollutants[7,8,9,10], water 59 being by far the most deleterious element” is incomplete, I’m not sure exactly what the authors are intending to communicate here.

answer

This part of the introduction has been modified to be clearer (Lines 57-60).

---

## [Decision Letter · Decision Letter 1]

28 Oct 2021

Long term conservation of DNA at ambient temperature. Implications for DNA data storage

PONE-D-21-23797R1

Dear Dr. Bonnet,

We’re pleased to inform you that your manuscript has been judged scientifically suitable for publication and will be formally accepted for publication once it meets all outstanding technical requirements.

Kind regards,

Jian Xu, Ph.D.

Academic Editor

PLOS ONE

Additional Editor Comments (optional):

Reviewers' comments:

Reviewer's Responses to Questions

**Comments to the Author**

1. If the authors have adequately addressed your comments raised in a previous round of review and you feel that this manuscript is now acceptable for publication, you may indicate that here to bypass the “Comments to the Author” section, enter your conflict of interest statement in the “Confidential to Editor” section, and submit your "Accept" recommendation.

Reviewer #2: All comments have been addressed

Reviewer #3: All comments have been addressed

2. Is the manuscript technically sound, and do the data support the conclusions?

Reviewer #2: Yes

Reviewer #3: Yes

3. Has the statistical analysis been performed appropriately and rigorously? 

Reviewer #2: Yes

Reviewer #3: Yes

4. Have the authors made all data underlying the findings in their manuscript fully available?

Reviewer #2: Yes

Reviewer #3: Yes

5. Is the manuscript presented in an intelligible fashion and written in standard English?

Reviewer #2: Yes

Reviewer #3: Yes

6. Review Comments to the Author

Reviewer #2: The present work is an excellent study and is now ready to be published in PLOS one. The authors have carefully read all suggestions, have answered each point, and have replied to the suggested changes.

Reviewer #3: The authors have greatly clarified their work. I find the revised work sufficient and greatly improved from their initial submission.

I recommend the work for publication.

7. PLOS authors have the option to publish the peer review history of their article (what does this mean?). If published, this will include your full peer review and any attached files.

Reviewer #2: No

Reviewer #3: No

---

## [Editor Report · Acceptance letter]

2 Nov 2021

PONE-D-21-23797R1 

Long term conservation of DNA at ambient temperature. Implications for DNA data storage. 

Dear Dr. Bonnet:

I'm pleased to inform you that your manuscript has been deemed suitable for publication in PLOS ONE. Congratulations! Your manuscript is now with our production department. 

Kind regards, 

on behalf of

Dr. Jian Xu 

Academic Editor

PLOS ONE